# The preponderance of nonsynonymous A-to-I RNA editing in coleoids is nonadaptive

Daohan Jiang [1] & Jianzhi Zhang [1]*

A-to-I editing enzymatically converts the base adenosine (A) in RNA molecules to inosine (I), which is recognized as guanine (G) in translation. Exceptionally abundant A-to-I editing was recently discovered in the neural tissues of coleoids (octopuses, squids, and cuttlefishes), with a greater fraction of nonsynonymous sites than synonymous sites subject to high levels of editing. Although this phenomenon is thought to indicate widespread adaptive editing, its potential advantage is unknown. Here we propose an alternative, nonadaptive explanation. Specifically, increasing the cellular editing activity permits some otherwise harmful G-to-A nonsynonymous substitutions, because the As are edited to Is at sufficiently high levels. These high editing levels are constrained upon substitutions, resulting in the predominance of nonsynonymous editing at highly edited sites. Our evidence for this explanation suggests that the prevalent nonsynonymous editing in coleoids is generally nonadaptive, as in species with much lower editing activities.

[1] Department of Ecology and Evolutionary Biology, University of Michigan, Ann Arbor, Michigan, USA. *email: jianzhi@umich.edu

RNA editing refers to a variety of posttranscriptional alterations of RNA molecules, including chemical modifications as well as insertions and deletions of nucleotides, but excluding RNA processing events such as splicing, capping, and polyadenylation[1,2]. Transcriptome-wide profiling of each type of RNA editing and understanding its biochemical and physiological functions are a major task of molecular and genome biology, and have seen a rapid progress in the last decade[3–6]. Among over 100 different types of RNA editing, adenosine (A)-to-inosine (I) editing of RNAs transcribed from animal nuclear genomes is arguably best studied[7–9]. The A-to-I conversion is catalyzed by a family of adenosine deaminase acting on RNA (ADAR) and the resultant I is recognized as guanine (G) in translation. For simplicity, we refer to A-to-I editing as A-to-G editing hereafter. If the editing takes place in protein-coding regions, it could be either nonsynonymous (also known as recoding) or synonymous, depending on whether the encoded amino acid is altered or not. A-to-G editing has been reported in multiple animal phyla[10,11], such as many vertebrates[10,12–17], as well as fruit flies[18–23], cephalopods[24–27], nematodes[28,29], and cnidarians[30]. Although an editing mechanism could emerge by chance and become fixed by genetic drift[31], studies of functional consequences of a handful of A-to-G recoding events led to the initial belief that recoding offers an "extreme advantage[32]," because disrupting recoding could be lethal[33]. This view has been challenged in the last few years by transcriptome-wide analysis of RNA editing. Specifically, there is a long tradition in molecular evolutionary genetics to compare the rate of synonymous nucleotide substitution ($d_S$) with that of nonsynonymous substitution ($d_N$) in protein-coding DNA sequence evolution. As synonymous changes are presumably neutral, while nonsynonymous changes may or may not be neutral, an observation of $d_N > d_S$ indicates overall positive selection promoting beneficial nonsynonymous substitutions, whereas $d_N < d_S$ indicates overall purifying selection hindering deleterious nonsynonymous substitutions. Although RNA editing is a molecular phenotype, similar comparisons between synonymous and nonsynonymous editing can be made[34]. For instance, in humans, the fraction of sites subject to nonsynonymous editing is lower than that subject to synonymous editing and the editing level (i.e., the proportion of RNA molecules edited at a site) is also lower for nonsynonymous than synonymous editing[34]. These patterns suggest that nonsynonymous editing is generally deleterious and is selectively removed and/or suppressed when compared with synonymous editing, which is presumably inconsequential to protein function. Therefore, most A-to-G coding RNA-editing events appear to be nonadaptive and are probably attributable to cellular errors resulting from ADARs' limited specificity[34]. This conclusion is compatible with the fact that only a handful of editing events have known functions[33], and that only 1.8% of ~2000 human coding RNA-editing events are shared with mouse[35,36].

The trend, however, is drastically different in coleoid cephalopods, which include octopuses, squids, and cuttlefishes. Tens of thousands of coding A-to-G editing events, including a considerable proportion of recoding, have been identified in the neural tissues of coleoids[25,27]. In particular, the frequency of nonsynonymous sites subject to high levels of editing exceeds that of synonymous sites, leading to the inference that nonsynonymous editing has been promoted by positive selection and is generally advantageous in coleoids[25,27]. We will refer to this hypothesis as the adaptive hypothesis. Furthermore, because the high editing activity appears to be limited to their neural tissues, it was speculated that the extraordinary abundance of RNA editing in coleoids is related to their complex nervous system and behavior[24,25,27,37]. Nonetheless, with the exception of recoding of an octopus potassium channel that is associated with cold

adaptation[26], no benefit of the widespread editing is known in coleoids. Here we propose and provide evidence for an alternative, nonadaptive explanation of the preponderance of highly edited nonsynonymous sites in coleoids.

## Results

**A nonadaptive hypothesis and its predictions.** Let us consider a genomic position in a coding region that is currently occupied by G and does not accept A (see top row in Fig. 1a). As the editing activity in the species rises, a G-to-A mutation at the site may become neutral and fixed if the resultant A is edited back to G in a sufficiently large proportion of mRNA molecules (see middle row in Fig. 1a). Upon the G-to-A substitution, the high editing level at the site will be selectively maintained, because it is G rather than A that is permissible at the mRNA level. As the above situation applies only to nonsynonymous G-to-A substitutions and the coupled nonsynonymous A-to-G editing, it inflates the number of nonsynonymous editing sites and nonsynonymous editing levels relative to the corresponding synonymous values. Although here the nonsynonymous editing has permitted the fixation of the otherwise deleterious G-to-A mutation, the derived genotype with a genomic A that is highly edited is no fitter than the original genotype with a genomic G. Thus, the editing is nonadaptive. We assumed in the above scenario that the editing level is so high that the otherwise deleterious G-to-A mutation becomes neutral. It is also possible that the editing level is not high enough, rendering the G-to-A mutation slightly deleterious (see bottom row in Fig. 1a). A slightly deleterious mutation may nevertheless get fixed and the editing level may be selectively increased in subsequent evolution. Even under this scenario, there is no net fitness gain from the original genotype with a genomic G to the derived genotype with a genomic A that is highly edited. We refer to the above nonadaptive model including both of the described scenarios as the harm-permitting model, because RNA editing permits the fixation of otherwise harmful mutations. Although the possibility of harm-permitting by RNA editing has been proposed multiple times[31,38–40], especially regarding the editing of organelle transcriptomes, empirical evidence that it is entirely or primarily responsible for creating "adaptive signals" of RNA editing is lacking.

Given the exceptionally high editing activity in coleoid neural tissues[25,27], we hypothesize that the reported preponderance of nonsynonymous editing is explained by the harm-permitting model and is nonadaptive. To test this hypothesis, we divide nonsynonymous editing into two categories: restorative and diversifying[41]. Restorative editing converts the amino acid state back to an ancestral state (Fig. 1b), whereas diversifying editing converts the amino acid state to a non-ancestral state (Fig. 1c). As restorative editing but not diversifying editing can confer a harm-permitting effect, our hypothesis predicts that the reported preponderance of nonsynonymous editing in coleoids is attributable to restorative but not diversifying editing. In particular, we predict that (i) the frequency of sites edited is greater for restorative ($F_R$) than synonymous ($F_S$) editing, and that (ii) the median editing level is higher for restorative ($L_R$) than synonymous ($L_S$) editing. It further predicts that (iii) the frequency of sites edited is no greater for diversifying ($F_D$) than synonymous ($F_S$) editing, and that (iv) the median editing level is no higher for diversifying ($L_D$) than synonymous ($L_S$) editing. By contrast, the adaptive hypothesis does not have specific predictions about $F_R$ and $L_R$, but predicts that $F_D$ and $L_D$ are respectively greater than $F_S$ and $L_S$. It is noteworthy that although only restorative editing can be harm-permitting, not all restorative editing is necessarily harm-permitting. For instance, the restorative editing would be neutral if it restores a neutral G-to-A substitution.

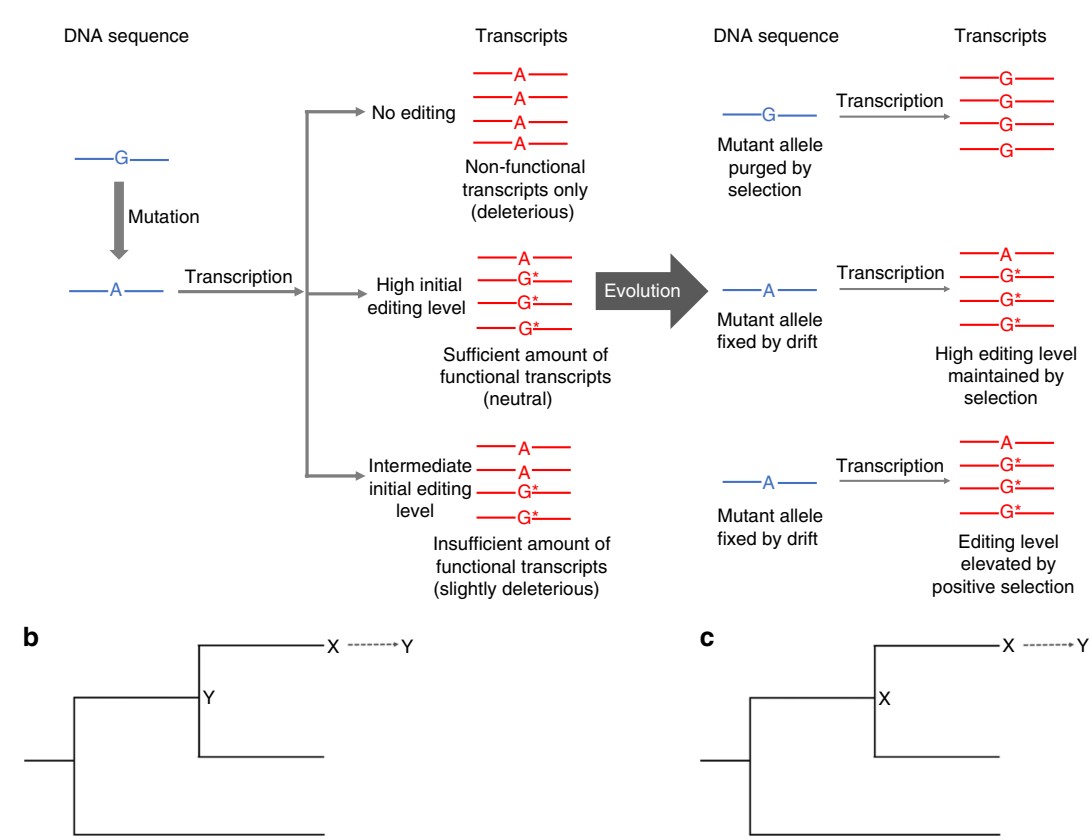

**Fig. 1** The harm-permitting model and a strategy to detect the harm-permitting effect. **a** The harm-permitting effect of nonsynonymous editing. The top row shows that, when a nonsynonymous A site is not edited (or is subject to a low level of editing), a G-to-A mutation at the site is too deleterious to get fixed. The middle row shows that, when the site is highly edited, the G-to-A mutation becomes neutral and is fixed by genetic drift. The high editing level is then selectively constrained. The bottom row shows that, when the editing level of the site is intermediate, the G-to-A mutation is slightly deleterious and fixed by genetic drift. The editing level may be further elevated by positive selection (or maintained by negative selection). Despite the relatively high nonsynonymous editing levels in the middle and bottom rows, no adaptation (i.e., no net increase in fitness) occurred when the final genotype is compared with the original genotype. DNA is shown in blue, whereas RNA is in red. Post-edited nucleotides are marked with stars. **b** Restorative editing restores an ancestral amino acid state lost upon an amino acid substitution, which may have occurred in the exterior branch as shown here or in an earlier branch. In other words, the post-editing state is identical to an ancestral pre-editing state. **c** Diversifying editing creates an amino acid state that differs from pre-editing states in a set of ancestors considered. Although only the state of one ancestor is shown here, the states of multiple ancestors may be considered. In **b** and **c**, X and Y represent different amino acid states, whereas the arrow shows the effect of editing. Restorative but not diversifying editing can confer a harm-permitting effect.

**Patterns of restorative and diversifying editing.** To test the nonadaptive hypothesis, we analyzed the published neural transcriptomes of six mollusk species[27], whose phylogenetic relationships are depicted in Fig. 2a. Among them, the four coleoids have widespread coding A-to-G editing in neural tissues, whereas the two outgroups have substantially fewer editing sites[27].

We identified 3587 one-to-one orthologous genes in these six species and inferred ancestral coding sequences at all interior nodes of the species tree (Fig. 2a). We regarded a nonsynonymous editing event in an exterior node of the tree that modifies the amino acid state from X to Y as restorative if the inferred genomic sequence-based amino acid state is Y at any node of the tree that is ancestral to the focal exterior node (Fig. 1b; also see Methods), or diversifying if Y is not present at any node of the tree that is ancestral to the focal exterior node (Fig. 1c). It is worth noting that these definitions are based on amino acid states and are applied to nonsynonymous editing only. Synonymous editing is presumably neutral, so need not be separated into restorative and diversifying editing. Furthermore, separating synonymous editing into the two categories would be less accurate because of lower reliabilities in inferring ancestral sequences at synonymous sites. Of the two categories of nonsynonymous editing sites, the

number of diversifying editing sites is 8.4–13.9 times that of restorative editing sites in the four coleoids (Supplementary Table 1).

In each of the four coleoids, $F_R$ and $L_R$ are significantly greater than $F_S$ (Fig. 2b) and $L_S$ (Fig. 2c), respectively. By contrast, $F_D$ is significantly smaller than $F_S$ (Fig. 2b), whereas $L_D$ is not significantly different from $L_S$ (Fig. 2c). These results confirm all four predictions of the nonadaptive hypothesis and are at odds with the predictions of the adaptive hypothesis, strongly suggesting that the preponderance of nonsynonymous editing in coleoids is explained by the harm-permitting model and is nonadaptive. Figure 2c shows that, although $L_R$ is significantly higher than $L_S$ in each coleoid, it is lower than 2.5%. One might ask whether such low median levels of restorative editing can be harm-permitting. As mentioned, not all restorative editing is necessarily harm-permitting, which could explain why $L_R$ is not particularly high. Nevertheless, Fig. 2c reveals a larger fraction of restorative editing than synonymous editing with appreciable editing levels. For example, in the squid, 33.37% and 13.31% of restorative editing sites but only 22.97% and 6.74% of synonymous editing sites have editing levels >5% and >20%, respectively. Depending on the harm of the G-to-A mutation

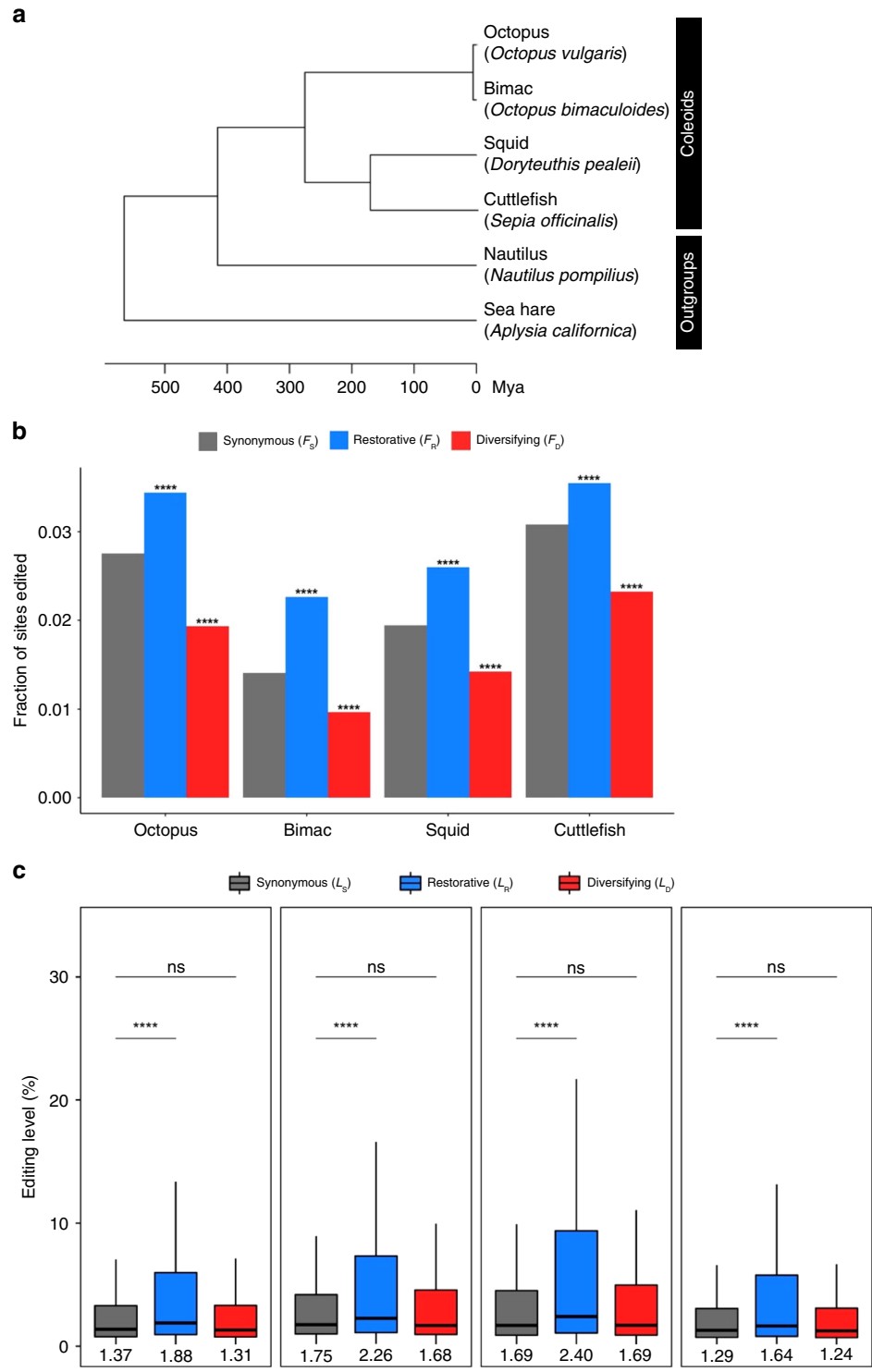

**Fig. 2** Comparison of restorative and diversifying editing with synonymous editing in coleoids. **a** The phylogenetic relationship of the six mollusks studied here. Branch lengths represent divergence times based on the mid-points of the divergence time ranges in a previous study[27]. **b** Frequencies of sites with synonymous ($F_S$), restorative ($F_R$), and diversifying ($F_D$) editing, respectively, in each of the four coleoids. A significant difference between $F_S$ and $F_R$ (or $F_D$) is indicated by stars above the bin of $F_R$ (or $F_D$) (*$P < 0.05$; **$P < 0.01$; ***$P < 0.001$; ****$P < 0.0001$; ns, not significant; $\chi^2$-test). **c** Synonymous ($L_S$), restorative ($L_R$), and diversifying ($L_D$) editing levels in each of the four coleoids. The lower and upper edges of a box represent the first ($qu_1$) and third quartiles ($qu_3$), respectively, the horizontal line inside the box indicates the median (md), and the whiskers extend to the most extreme values inside inner fences, md ± 1.5($qu_3 - qu_1$). The median editing levels are also given below the corresponding boxes. A significant difference between $L_S$ and $L_R$ (or $L_D$) is indicated by stars (*$P < 0.05$; **$P < 0.01$; ***$P < 0.001$; ****$P < 0.0001$; ns, not significant; Mann–Whitney $U$-test). Source data are provided as a Source Data file.

and the relative dominance of the A and G isoforms, these appreciable levels of A-to-G editing could substantially increase the fixation probability of the G-to-A mutation. It should also be noted that the harm-permitting hypothesis is proposed as an alternative to the adaptive hypothesis. If moderate levels of nonsynonymous editing could be beneficial as asserted by the adaptive hypothesis, there is no reason why they could not be harm-permitting. Furthermore, the general trend of $L_R > L_S$ and $L_D < L_S$ supports the harm-permitting hypothesis relative to the adaptive hypothesis.

To examine the robustness of our results, we conducted four additional analyses. First, we respectively examined editing sites that are specific to each of the four coleoids, because species-specific editing events have similar evolutionary ages, allowing fairer comparisons. The results obtained are highly similar to those in Fig. 2 and are robust to potential misidentifications of species-specific editing (Supplementary Fig. 1). Second, we probed editing events identified from individual tissues in bimac. $F_R > F_S$ and $F_D < F_S$ hold across tissues, but editing level comparisons are mostly nonsignificant, likely due to the reduced statistical power as a result of decreased sample sizes (Supplementary Table 2). Third, because editing levels of neighboring editing sites may be co-affected by a mutation, which would reduce the statistical power in comparing synonymous with nonsynonymous editing sites, we compared synonymous editing sites in one half of the gene set with nonsynonymous editing sites in the other half. Specifically, we ranked all genes by the $d_N/d_S$ ratio between octopus and squid orthologs, and respectively grouped genes with odd ranks into bin 1 and those with even ranks into bin 2. We then compared synonymous editing in bin 1 with nonsynonymous editing in bin 2, as well as synonymous editing in bin 2 with nonsynonymous editing in bin 1. The results (Supplementary Fig. 2) are similar to those obtained from all editing sites (Fig. 2). Fourth, we respectively investigated $F_R/F_S$ and $F_D/F_S$ in five editing level ranges (0–20%, 20–40%, 40–60%, 60–80%, and 80–100%) in each coleoid (Supplementary Fig. 3). Both $F_R/F_S$ and $F_D/F_S$ generally increase with the editing level. Although $F_R/F_S$ almost always exceeds 1, $F_D/F_S$ is smaller than 1, except when the editing level exceeds 60%. It is important to stress that only a few percent of diversifying editing sites in a coleoid fall in this editing level range (Supplementary Table 3), suggesting that the vast majority of diversifying editing is nonadaptive (see below for quantitative estimates).

**Accelerated nonsynonymous G-to-A substitutions**. The harm-permitting model further predicts that the rate of nonsynonymous G-to-A substitution relative to that of synonymous G-to-A substitution ($d_N/d_S$ for G-to-A) should be elevated, because the high editing activity renders some otherwise deleterious nonsynonymous G-to-A mutations acceptable. Furthermore, this elevation should be particularly pronounced in genes exclusively expressed in neural tissues but not in genes unexpressed in neural tissues, because the high editing activity is so far observed only in neural tissues[25,27]. However, because only bimac and squid have available RNA-sequencing data from several non-neural tissues and because genes unexpressed in neural tissues are not in the transcript sequence data of the octopus and cuttlefish, and hence are excluded from our alignments, we had to define two groups of genes with relatively high and relatively low specificities in neural expression, respectively. The genes with high neural expression specificities are expressed exclusively in neural tissues in the bimac or squid, whereas those with low neural expression specificities are expressed in both neural and non-neural tissues in both the bimac and squid. The harm-permitting model predicts that $d_N/d_S$ for G-to-A is greater for genes with relatively high

neural expression specificities than for those of relatively low neutral expression specificities. As the harm-permitting effect is present only when a G-to-A mutation at a site is deleterious without editing, we focused on nonsynonymous sites that are conserved in the two outgroup species (i.e., nautilus, sea hare, and the immediately ancestral node of the focal species share the same pre-editing state) to increase the sensitivity of our test. Furthermore, the elevation in $d_N/d_S$ should be specific to G-to-A changes, because the potential harms of other changes such as C/T-to-A and G-to-C/T cannot be alleviated by A-to-G editing.

To this end, we considered all six branches descendent from the common ancestor of the four coleoids. We computed $d_N$ and $d_S$ of each of these branches using the extant and inferred ancestral sequences, and then calculated $d_N/d_S$ by dividing the total $d_N$ by the total $d_S$ of these branches. In support of our prediction, $d_N/d_S$ for G-to-A changes is greater for genes of relatively high neural expression specificities than those of relatively low specificities (Fig. 3). By respectively bootstrapping the two groups of genes 200 times, we found that the above difference is statistically significant ($P = 0.015$). By contrast, no significant difference in $d_N/d_S$ exists between the two groups of genes when C/T-to-A changes or G-to-C/T changes are considered (Fig. 3). It is noteworthy that $d_N/d_S < 1$ in all cases in Fig. 3, consistent with the harm-permitting model that does not involve positive selection.

**The potential benefit of shared editing among species**. It has been suggested that shared editing among multiple species is likely beneficial, because otherwise the editing status is unlikely to be evolutionarily conserved[36]. In support of this suggestion was the finding that, even in mammals, where most nonsynonymous editing appears neutral or deleterious, the frequency of conserved sites subject to nonsynonymous editing in both human and mouse significantly exceeds the frequency of conserved sites subject to synonymous editing in both species[36]. A similar phenomenon was reported in fruit flies[23]. In coleoids, a sizable fraction of nonsynonymous editing is shared by at least two species and highly edited sites tend to be shared[27]. To understand the potential evolutionary forces maintaining RNA editing at specific sites across multiple coleoids, we analyzed editing shared by a clade of two or more species.

A nonsynonymous editing event shared by a clade of species that modifies the amino acid state from X to Y is considered

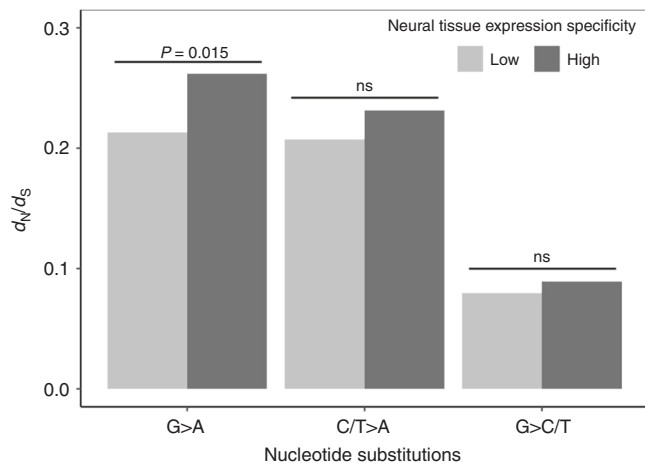

**Fig. 3** Coleoid nonsynonymous to synonymous substitution rate ratios ($d_N/d_S$) for various nucleotide changes. The P-value is based on 200 bootstrap samples; ns, not significant. Source data are provided as a Source Data file.

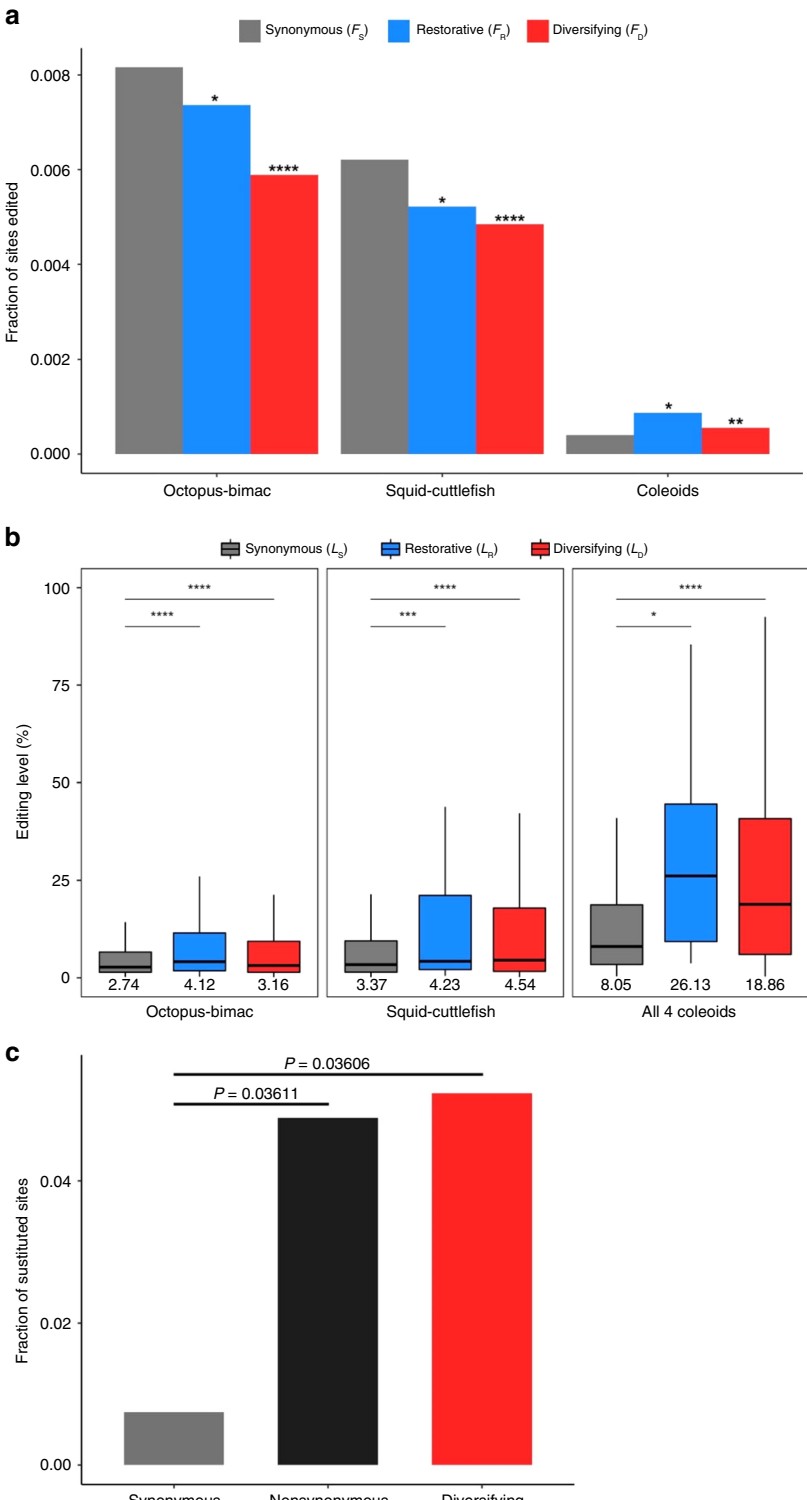

**Fig. 4 Patterns of shared editing in coleoids. a** Frequencies of sites with shared synonymous ($F_S$), restorative ($F_R$), and diversifying ($F_D$) editing, respectively, among sites shared between the octopus and bimac, between the squid and cuttlefish, and among all four coleoids, respectively. A significant difference between $F_S$ and $F_R$ (or $F_D$) is indicated by stars above the bin of $F_R$ (or $F_D$) (*$P < 0.05$; **$P < 0.01$; ***$P < 0.001$; ****$P < 0.0001$; ns, not significant; $\chi^2$-test). **b** Synonymous ($L_S$), restorative ($L_R$), and diversifying ($L_D$) editing levels among the three sets of shared editing sites. The lower and upper edges of a box represent the first (qu$_1$) and third quartiles (qu$_3$), respectively, the horizontal line inside the box indicates the median (md), and the whiskers extend to the most extreme values inside inner fences, md ± 1.5(qu$_3$ − qu$_1$). The median editing levels are also given below the corresponding boxes. A significant difference between $L_S$ and $L_R$ (or $L_D$) is indicated by stars (*$P < 0.05$; **$P < 0.01$; ***$P < 0.001$; ****$P < 0.0001$; ns, not significant; Mann–Whitney $U$-test). **c** Fraction of sites edited in the common ancestor of the four coleoids that have a genomic G in a coleoid. $P$-values are based on two-tailed Fisher's exact test. Source data are provided as a Source Data file.

restorative if the inferred genomic sequence-based amino acid state is Y at any node of the tree that is ancestral to the most recent common ancestor of the clade, or diversifying if Y is not present at any of these ancestral nodes. In the study of shared editing, we considered the average editing level in the clade where the editing is shared. For editing sites shared between the octopus and bimac, and those shared between the squid and cuttlefish, $F_R$ and $F_D$ are both significantly smaller than $F_S$ (Fig. 4a). By contrast, $L_R$ and $L_D$ are both significantly greater than $L_S$ (Fig. 4b). For the subset of the above shared editing sites that are shared by all four coleoids, $F_D$ and $L_D$ are respectively significantly greater than $F_S$ (Fig. 4a) and $L_S$ (Fig. 4b), so are $F_R$ (Fig. 4a) and $L_R$ (Fig. 4b). A significantly greater $F_D$ than $F_S$ for shared editing could be caused by (i) positive selection promoting the initial fixation of mutations that lead to nonsynonymous editing and/or (ii) purifying selection preventing the loss of presumably beneficial nonsynonymous editing; therefore, it is a clear indicator of adaptive nonsynonymous editing. A significantly greater $L_D$ than $L_S$ for shared editing could be caused by (i) positive selection promoting the increase of editing levels of presumably beneficial nonsynonymous editing, (ii) purifying selection preventing the decrease of editing levels of presumably beneficial nonsynonymous editing, (iii) purifying selection preferentially preventing the loss of high-level nonsynonymous editing presumably because high editing levels are associated with larger benefits than low editing levels, and/or (iv) positive selection preferentially promoting the loss of low-level nonsynonymous editing, probably because an A-to-G substitution is favored at an edited site, especially when the editing level is low. Regardless, a significantly greater $L_D$ over $L_S$ also indicates adaptive nonsynonymous editing. Hence, nonsynonymous editing shared by all four coleoids show strong and consistent adaptive signals, suggesting that a large fraction is adaptive. In comparison, nonsynonymous editing shared between the octopus and bimac, and that shared between the squid and cuttlefish exhibit some but not all signs of adaptation, and the adaptive signals are much weaker, suggesting that only a smaller fraction is adaptive.

As most nonsynonymous editing is species-specific (Supplementary Table 1), the above finding is not inconsistent with the analysis of individual species revealing the nonadaptive nature of most editing events. We estimated that, of species-specific diversifying editing sites, 0.47%, 0.52%, 1.12%, and 0.40% are adaptive in the octopus, bimac, squid, and cuttlefish, respectively (see Methods). Similarly, 1.65%, 1.42%, 8.31%, and 4.95% of shared diversifying editing sites are adaptive in the four coleoids, respectively. Taken together, 0.75%, 0.98%, 1.90%, and 1.00% of diversifying editing sites are adaptive in the four coleoids, respectively.

What is the general benefit of the shared editing that shows adaptive signals? Two hypotheses exist. First, editing may be beneficial because of the intra-organism protein diversity created[25,27,32,42]. That is, editing allows the existence of two protein isoforms per edited site in an organism, which may confer a higher fitness, analogous to heterozygote advantage at polymorphic sites. Alternatively, editing offers a new isoform that may be simply fitter than the unedited isoform. In this latter hypothesis, the benefit of editing is comparable to that of a nucleotide substitution. To distinguish between these two hypotheses, we focused on sites that are edited in at least three of the four coleoids, because editing should have existed at these sites in the common ancestor of the four species according to the parsimony principle (Fig. 2a). We then estimated the frequency of replacement of editing with an A-to-G substitution in any of the four species. Such replacements are expected to be more or less neutral for synonymous editing. For nonsynonymous editing,

such replacements are deleterious under the first hypothesis due to the loss of protein diversity but are neutral under the second hypothesis. Hence, the first hypothesis predicts a lower frequency of such replacements for nonsynonymous editing than synonymous editing, whereas the second hypothesis predicts equal frequencies of such replacements for synonymous and nonsynonymous editing.

Interestingly, the frequency of such replacements for nonsynonymous editing is significantly greater than that for synonymous editing in a two-tailed Fisher's exact test (Fig. 4c and Supplementary Table 4). Because it is the shared diversifying editing for which the nature of the benefit is in question, we restricted the analysis to diversifying editing only, but obtained a similar result (Fig. 4c and Supplementary Table 4). It is noteworthy that no synonymous or nonsynonymous editing was found to be replaced with an A-to-C/T substitution among this set of sites (Supplementary Table 4). Our finding suggests that, if anything, nonsynonymous editing is more likely to be replaced with an A-to-G substitution than is synonymous editing, probably because having a genomic G is superior to having a genomic A that cannot be edited to G in all mRNA molecules. In other words, our results reject the first hypothesis and suggest that the nature of the benefit of adaptive A-to-G editing is similar to that of the same nucleotide substitution, although the size of benefit from the former is smaller than that from the latter. Furthermore, the finding in Fig. 4c suggests that the significantly greater $F_D$ than $F_S$ for editing shared among all four coleoids is better explained by positive selection promoting the initial fixation of mutations that led to beneficial nonsynonymous editing than purifying selection preventing the loss of beneficial nonsynonymous editing.

## Discussion

The recent discovery of the preponderance of nonsynonymous A-to-G RNA editing among highly edited sites in coleoid neural tissues led to the assertion of widespread adaptive editing in these organisms, but the potential benefits of the editing are unknown. In this work, we proposed an alternative, nonadaptive explanation. Our reanalysis of published transcriptome data from four coleoids and two outgroup species lends strong support to the nonadaptive hypothesis. Combined with previous findings from other species, the new finding suggests a generally nonadaptive nature of coding A-to-G editing among animals. As explained in the harm-permitting model, nonadaptive editing such as some restorative editing, may, however, be selectively protected (middle row in Fig. 1a) or even promoted (bottom row in Fig. 1a). Although such editing events likely originated as molecular errors due to ADARs' limited target specificity, they are no longer errors today. The fact that a nonadaptive feature can nevertheless be under purifying selection or even be positively selected is well known in evolutionary biology[40,43].

In the harm-permitting model, A-to-G editing permits the fixation of otherwise deleterious G-to-A mutations and hence the editing is nonadaptive. In theory, it is also possible that A-to-G editing emerged in evolution after a G-to-A substitution at the same site. If the substitution is slightly deleterious, the editing would be slightly beneficial (i.e., compensatory). However, such sites have minimal contributions to $F_R$ and $L_R$, so this possibility does not alter our interpretation of the nonadaptive nature of restorative editing (see Methods).

The principle of our test of the nonadaptive hypothesis of RNA editing is similar to that of the test of the adaptive hypothesis, except that the new test requires a distinction between restorative and diversifying editing, which in turn depends on ancestral coding sequences inferred for the interior nodes of a phylogeny

(Fig. 1b, c). Although ancestral sequence inference is generally reliable, it is not expected to be 100% correct[44]. Will errors and potential biases in this inference bias our test? The answer is no. $F_R$ is the number of edited sites with an ancestral nonsynonymous G-to-A substitution divided by the total number of sites with an ancestral nonsynonymous G-to-A substitution. As our ancestral sequence inference is based on genomic sequences and is blind to RNA editing, any potential bias in estimating the number of sites with an ancestral G-to-A substitution is cancelled out in computing $F_R$. The same applies to $F_D$. Errors and potential biases in ancestral sequence inference only increase the stochastic errors of $F_R$ and $F_D$ estimates, reducing the statistical power in testing our hypothesis. Notwithstanding, the vast majority of our key statistical tests yielded significant results, suggesting that sufficient statistical power remains in these tests.

Although our study explains the preponderance of nonsynonymous editing in coleoids, we have not addressed a related question—why the editing activity was drastically elevated in neural tissues during coleoid evolution. A substantial rise in editing activity is expected to be harmful, because its effect is similar to inducing A-to-G mutations. Indeed, expression of the human ADAR2 gene in the budding yeast Saccharomyces cerevisiae, which does not naturally possess any ADAR gene, inhibits yeast growth because of ADAR2's RNA editing activity[45]. Our observation of a significantly lower $F_D$ than $F_S$ in every coleoid examined (Fig. 2b) strongly suggests that diversifying editing is generally deleterious and has been selectively purged. Hence, it is almost certain that the pervasive coding RNA editing was not the reason for the elevation of the editing activity in coleoids but its byproduct. Whatever the reason was, the relevant benefit must at least offset the harm from pervasive nonsynonymous editing, under the assumption that the evolutionary elevation of the editing activity was not due to genetic drift alone, because the population size of ancestral coleoids was probably not small. It is worth mentioning that a number of physiological functions have been proposed for A-to-G editing, including suppressing the proliferation of transposons[46], inhibiting viral replication[47], marking RNAs for degradation[32], marking RNAs to prevent innate immunity against self-RNAs[48,49], regulating alternative splicing[32], and modulating nuclear retention of RNAs[32]. As the primary physiological function of A-to-G editing is unknown, it is difficult to discern why the editing activity rose drastically in coleoids.

Similar to previous findings in mammals and flies[23,36], we observed some adaptive signals from nonsynonymous editing shared between species. Our additional analysis suggests that the benefit of these adaptive editing events does not lie in the protein diversity brought by editing, but lies in the superiority of the edited isoform to the unedited version. Furthermore, nonsynonymous editing is more likely than synonymous editing to be replaced with an A-to-G substitution, suggesting that the nature of the benefit of adaptive editing is similar to the corresponding nucleotide substitution but the extent of the benefit is smaller than that of the substitution. Thus, even when RNA editing is advantageous, the advantage does not rely on its characteristic of generating protein diversity; rather, editing appears to be a temporary solution that is eventually replaced by the more advantageous A-to-G substitution. This result contrasts the prevailing view about how coding RNA editing may be adaptive and further argues that coding sequence editing is unlikely the primary function of RNA editing.

Liscovitch-Brauer and colleagues[27] noted that flanking regions of sites edited in multiple species tend to be evolutionarily conserved and asserted that coleoids "use extensive RNA editing to diversify their neural proteome at the cost of limiting genomic sequence flexibility and evolution." Contrary to this interpretation, nonsynonymous editing of the common ancestor of coleoids is more likely than synonymous editing to be replaced with an A-to-G substitution. That is, an A-to-G substitution is preferred over A-to-G editing even when the editing is beneficial. We believe that the observation prompting Liscovitch-Brauer et al.'s[27] erroneous conclusion is caused by an ascertainment bias. Specifically, because of the various requirements for a site to be edited, such as specific flanking sequences[27] and secondary structures[50], a shared editing site by definition satisfies these requirements in its neighborhood in multiple species. Thus, the site is expected to show a higher interspecific similarity in flanking sequences than a randomly picked site, regardless of whether the editing is shared because of selective constraints or not. The same ascertainment bias occurs in the comparison of intraspecific polymorphisms of flanking sequences between shared editing sites and random sites. In particular, given the flanking sequence requirement for editing, an edited site with a lower flanking sequence polymorphism is expected to be edited in a greater percentage of individuals in the species. Hence, provided that a site is found to be edited in multiple species when only one individual is examined per species, the polymorphism is expected to be low irrespective of the presence/absence of selective constraints on the editing.

The nonadaptive hypothesis we proposed is based on the harm-permitting effect of high levels of editing, which inflates the frequency and level of restorative editing, relative to those of synonymous editing. As previous comparisons of synonymous and nonsynonymous editing in non-coleoid species never considered this effect, one wonders whether their conclusions are still valid. Ignoring the harm-permitting effect renders conclusions of nonadaptive editing more conservative. Hence, such conclusions should still hold. For claims of adaptive editing that are based on comparisons between synonymous and nonsynonymous editing frequencies and levels, a reanalysis taking into account the harm-permitting effect is warranted. In other words, a significantly greater $F_D$ than $F_S$ and/or a significantly greater $L_D$ than $L_S$ are required to demonstrate positive selection promoting nonsynonymous editing. This is especially true to the group of fungi that show pervasive A-to-G editing as in coleoids[51–53].

It is worth mentioning that transcriptome-wide analyses of several other types of RNA editing such as C-to-U editing[54] and m[6]A modification (methylation of A at the nitrogen-6 position)[55] also suggest that most editing events are nonadaptive. In addition, variations in several steps of RNA production and processing such as alternative transcriptional initiation[56], alternative splicing[57], and alternative polyadenylation[58] have been shown to be largely molecular errors. Similarly, it is plausible that variations in the translational process such as stop-codon read-through[59] and events of posttranslational modifications such as phosphorylation[60] and glycosylation[61] are primarily manifestations of molecular errors. Whether it is generally true that phenotypic variations at the molecular level are less likely to be adaptive than those at the cellular, tissue, organ, and organismal levels is worth exploration[62].

## Methods

**Transcriptomes, editing sites, and ancestral sequences**. The transcriptomes of six mollusk species and the list of A-to-I editing sites in the four coleoid species were previously published[27]. We extracted coding sequences from the previously assembled transcriptomes[27] on the basis of the annotations in the dataset. In some genes, we observed stop codons occurring upstream of the last three nucleotides of the annotated coding sequence, possibly due to erroneous inclusions of 3′-untranslated regions. We therefore removed nucleotides downstream of the first stop codon in these sequences. All but one A-to-G editing site in the data are upstream of the first stop codons, suggesting that these annotation errors barely influenced the previous analysis of RNA editing. If a gene appeared more than once in the original dataset for a species, only the longest sequence was retained in our analyses.

Orthologous genes among the six mollusks were previously identified[27] and a total of 3587 genes have orthologs in all 6 species and contain at least 1 A-to-G editing site in at least 1 coleoid. We first made a protein sequence alignment of orthologous sequences using Clustal Omega[63] and then generated a coding sequence alignment of these genes using PAL2NAL[64]. Ancestral sequences were inferred using the codeml program in PAML4[65] under default parameters and the best joint inferences of all interior nodes were used in subsequence analyses. The unrooted topology of the tree in Fig. 2a was used in ancestral sequence inference. Subsequent analyses used in-house Perl scripts.

All reported editing sites in the 3587 genes[27] were included in our analyses, unless otherwise noted. Although some editing sites may be sequencing errors, the probability of error is expected to be low given the tiny amount of other types of DNA–RNA mismatches observed[27].

**Restorative and diversifying editing.** The tree in Fig. 2a shows three interior nodes ancestral to each coleoid species. A coding A site in a coleoid is considered a potential site for restorative editing if changing the A to G is nonsynonymous and if the corresponding amino acid after the change becomes identical to the amino acid state at any one of the three ancestral nodes. A potential site for restorative editing becomes a restorative editing site if it is edited in the focal species. By definition, $F_R$ is the number of sites with restorative editing divided by the number of potential sites for restorative editing, whereas $L_R$ is the median editing level at restorative editing sites. A coding A site in a focal species is considered a potential site for diversifying editing if changing the A to G is nonsynonymous and if the corresponding amino acid after the change differs from all amino acid states of the three ancestral nodes. A potential site for diversifying editing becomes a diversifying editing site if it is edited in the focal species. By definition, $F_D$ is the number of sites with diversifying editing divided by the number of potential sites for diversifying editing, whereas $L_D$ is the median editing level at diversifying editing sites. $F_S$ is the number of sites with synonymous editing divided by the number of A sites where A-to-G editing would be synonymous, whereas $L_S$ is the median editing level at synonymous editing sites[34]. Although the comparison between $F_R$ (or $F_D$) and $F_S$, and that between $L_R$ (or $L_D$) and $L_S$ are not entirely independent from each other, each comparison is fair.

An editing event is considered to be shared by a clade of two or more species if the event occurs in all species of the clade in the tree of Fig. 2a and all of these species have the same pre- and post-editing amino acid states. In studying shared editing by a clade, we followed the above procedure in distinguishing restorative from diversifying editing, except that we considered all interior nodes ancestral to the most recent common ancestor of the clade instead of all interior nodes ancestral to one species.

**Comparing median editing levels.** When the mRNA concentration is low, RNA editing cannot be detected unless the editing level is sufficiently high. This bias would make the median editing level appear higher in weakly expressed genes than strongly expressed genes even when no such difference actually exists. To alleviate this bias, we considered only those sites that are covered by at least 400 RNA-sequencing reads when comparing median editing levels. Nevertheless, the bias does not affect the comparison between synonymous and nonsynonymous editing, because their detectabilities are equally influenced by the gene expression level. For a shared editing site, the average editing level and average read number of all species in the focal clade are used to represent the site. We did not apply editing level cutoffs in the comparison of editing levels of different sites due to potential biases that may arise.

**Proportion of diversifying editing that is adaptive.** Under the presumption that the excess of $F_D$ over $F_S$ represents adaptive editing, we calculated $F_D$ and $F_S$ in each of 10 editing level intervals (0–10%, 10–20%, till 90–100%). For each interval exhibiting $F_D > F_S$, the number of adaptive diversifying editing sites equals ADP = $N_D(1 - F_S/F_D)$, where $N_D$ is the number of diversifying editing sites in the interval. Summing up these ADP numbers yields the total number of diversifying editing sites that are adaptive.

**Contributions of compensatory editing to $F_R$ and $L_R$.** In the harm-permitting model, A-to-G editing permits the fixation of otherwise deleterious G-to-A mutations and hence the editing is nonadaptive. In theory, it is also possible that A-to-G editing emerged in evolution after a G-to-A substitution at the same site. If the substitution is slightly deleterious, the editing would be slightly beneficial (i.e., compensatory). For several reasons, such sites should contribute minimally to $F_R$ and $L_R$. First, the probability that the G-to-A substitution occurred in the most recent common ancestor of cephalopods (the top five species in Fig. 2a) or more recently is small, because it could occur at any time prior to the emergence of the editing at the site, which most likely took place when the cellular editing activity rose substantially in the branch immediately preceding the common ancestor of coleoids. Hence, the probability that the editing is classified as restorative is small and such compensatory events are unlikely to affect our analysis of restorative editing sites. Although such compensatory events are potentially included in diversifying editing sites we analyzed, diversifying editing still show lower editing frequencies and editing levels when compared with synonymous editing. Thus, our

interpretation that diversifying editing is overall under purifying selection remains valid. Furthermore, even for the minority of compensatory events that are classified as restorative, the impact is small. This is because deleterious G-to-A mutations that could get fixed without editing are presumably only slightly deleterious. Hence, the benefit of A-to-G editing at such sites is also presumably small such that their editing level may not be selectively raised or selectively maintained at high levels. More importantly, there will be a comparable number of slightly beneficial G-to-A substitutions followed by slightly deleterious A-to-G editing that are included in the category of restorative editing. The effects of these two groups of events are likely cancelled out.

**Reporting summary.** Further information on research design is available in the Nature Research Reporting Summary linked to this article.

## Data availability

The original data were previously published. All new results are in the manuscript including Supplementary Materials. The source data underlying Figs. 2–4 and Supplementary Figs. 1–3 are provided as a Source Data file. Intermediate results are available from the authors.

## Code availability

Custom code is available from the authors.

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

## Acknowledgements

We thank Zhen Liu and members of the Zhang lab for valuable comments. This work was supported by the U.S. National Institutes of Health research grant GM120093 to J.Z.

## Author contributions

J.Z. conceived the project and secured funding. D.J. compiled and analyzed the data. D.J. and J.Z. designed the analyses and wrote the paper.

## Competing interests

The authors declare no competing interests.
