## [Peer Review File · Nature Communications]

Reviewers' Comments:

Reviewer #1:

Remarks to the Author:

Review of "The preponderance of nonsynonymous A-to-I RNA editing in coleoids is nonadaptive" by Jiang & Zhang (NCOMMS-18-30384)

In the manuscript, Jiang & Zhang proposed a "harm-permitting model" of RNA-editing, which posits that A-to-I RNA editing permits the fixation of otherwise harmful G-to-A DNA mutations because I is interpreted as G during cellular activities. To test their model, the authors reanalyzed the RNA editing data generated by Dr. Eli Eisenberg and colleagues (Liscovitch-Brauer et al. 2017, Cell). The authors focused on 3587 one-to-one orthologous genes in six cephalopod species and inferred ancestral coding sequences at all interior nodes of the species trees. By doing this, the authors divided the recoding editing sites into "restorative" and "diversifying" editing sites. The authors found strong signals of the adaptation in the restorative recoding editing sites, and surprisingly, the authors found the diversifying recoding editing sites are overall non-adaptive. Consistent with the "harm-permitting model", the authors found elevated dN/dS values specific to G-to-A changes in the neuron specific genes, while such a pattern was not observed in the C/T-to-A or G-to-C/T changes.

Overall, the model is well conceived and the manuscript is very well written. The results are very inspiring and would be very interesting to the RNA Biology field. Therefore, I would be happy to recommend acceptance after the authors make some revision.

Specific comments:

- 1) The authors treat all the "restorative" editing sites as "harm-permitting". This might not be necessarily the case. Let us suppose two scenarios: 1) a G->A DNA mutation is deleterious but somehow get fixed, and then A-to-I editing occurs on the A site, whose restorative effect is to rescue the G->A deleterious effect; and 2) a G->A DNA mutation is deleterious and segregating in the population, and then A-to-I editing occurs on the derived A site, whose effect is to change the deleterious effect of G->A mutation into slightly deleterious or neutral. This reviewer thinks in these two scenarios the restorative effect of editing is different: the latter is harm-permitting; while the former is rescue, which might be defined as "adaptive" since it can increase the fitness after the fixation of the deleterious mutations. The authors might consider separate these two scenarios in their model (Fig. 1) and their data analysis.
- 2) The dN/dS analysis used by the authors and other previous studies is designed to detect the signature of natural selection on DNA mutations. The editing events can be actually treated as "phenotypes" since they are determined and influenced by cis-regulatory elements. The authors are encouraged to discuss about this point to make their results more influential.

Since the conclusion drawn by Jiang & Zhang is somewhat opposite to that of Liscovitch-Brauer et al. (2017, Cell), to be fair, the authors might consider doing the following analysis.

- 3) The authors inferred the ancestral state of the sites only for the nonsynonymous editing, and compare the editing density to the synonymous sites. Is it possible the authors also infer the restorative and diversifying synonymous sites and use those as control in the analysis?
- 4) The authors analyzed 3587 1-to-1 orthologous genes in the six species. How about the remaining genes? If the authors remove the sites that show signals of restorative editing and do dN/dS analysis at the genome-scale, what would the results look like?
- 5) The editing level is not considered in the analysis, can the authors provide similar plot as Fig. 5A and 5B in Liscovitch-Brauer et al. 2017, Cell?

6) The authors detect the signals of adaptation by pooling all the editing sites from different tissues. If the authors analyzed the data for each individual tissue, do the patterns change?

Reviewer #2:

Remarks to the Author:

In this study, the authors proposed that the widespread nonsynonymous RNA editing in coleoids is generally nonadaptive, which is in contrast to the conclusion of a previous study by Liscovitch-Brauer and colleagues. It is indeed interesting to investigate the evolutionary forces that shape the abundant RNA editing events in coleoids. However, I found that the analysis is not strong enough to allow the authors to draw such a conclusion.

Specific concerns:

There are not enough details about the parameters used to select genuine RNA editing sites used for analysis. The median editing levels of different groups of sites in several figures (Fig. 2C, Fig. 4B, Fig. S1B) is 1-2% (nearly equal to the sequencing error rate of NGS data), suggesting that most sites the authors used are not even really edited. Any conclusion based on such site lists is unreliable. The authors should require a minimum editing level (e.g. 5%) to select sites for analysis.

The authors proposed the harm-permitting model of RNA editing. In this model, they consider a genomic position in a coding region that is occupied by G and does not accept A. As the editing activity in the species rises, a G-to-A mutation at the site may become neutral and fixed by genetic drift if the resultant A is edited back to G in a sufficient proportion of mRNAs. In practice, most editing events are edited in neuronal tissues only and the unedited forms of mRNAs are present in the non-neuronal tissues. Thus the editing positions do accept A and the harm-permitting model is invalid for most of the editing positions.

Even in the neuronal tissues, the median editing level of restorative sites is only around 2% and most mRNAs are actually unedited (Fig. 2C). So the A allele is also accepted in the neuronal tissues. This observation does not support the harm-permitting model proposed by the authors.

Fig. 2C: The observed lower editing levels of diversifying sites are likely because these sites are newly evolved and have not been optimized for editing. To allow for a fair comparison between L_d and L_s , a matched evolutionary age between diversifying sites and control synonymous sites is needed. In fact, as shown in Fig. 4B, when using sites with similar evolutionary age, the authors found that L_d is higher than L_s , which supports the adaptive hypothesis.

It is known that editing sites tend to be clustered together. Conserved nonsynonymous sites typically have higher ADAR binding affinity and editing levels, thus the nearby synonymous sites tend to have higher editing levels also. This kind of confounding effect may affect the editing level comparison between synonymous and diversifying sites. So a more rigorous selection of matched synonymous sites is needed throughout the analysis.

Responses to the reviewers

We are grateful to the two reviewers for their valuable comments, which have helped to strengthen our manuscript. Below please find our point-to-point response.

Reviewer #1

Comment 1

In the manuscript, Jiang & Zhang proposed a “harm-permitting model” of RNA-editing, which posits that A-to-I RNA editing permits the fixation of otherwise harmful G-to-A DNA mutations because I is interpreted as G during cellular activities. To test their model, the authors reanalyzed the RNA editing data generated by Dr. Eli Eisenberg and colleagues (Liscovitch-Brauer et al. 2017, Cell). The authors focused on 3587 one-to-one orthologous genes in six cephalopod species and inferred ancestral coding sequences at all interior nodes of the species trees. By doing this, the authors divided the recoding editing sites into “restorative” and “diversifying” editing sites. The authors found strong signals of the adaptation in the restorative recoding editing sites, and surprisingly, the authors found the diversifying recoding editing sites are overall non-adaptive. Consistent with the “harm-permitting model”, the authors found elevated dN/dS values specific to G-to-A changes in the neuron specific genes, while such a pattern was not observed in the C/T-to-A or G-to-C/T changes.

Overall, the model is well conceived and the manuscript is very well written. The results are very inspiring and would be very interesting to the RNA Biology field. Therefore, I would be happy to recommend acceptance after the authors make some revision.

Response

We thank the reviewer for the overall positive evaluation.

Comment 2

The authors treat all the “restorative” editing sites as “harm-permitting”. This might not be necessarily the case. Let us suppose two scenarios: 1) a G->A DNA mutation is deleterious but somehow get fixed, and then A-to-I editing occurs on the A site, whose restorative effect is to rescue the G->A deleterious effect; and 2) a G->A DNA mutation is deleterious and segregating in the population, and then A-to-I editing occurs on the derived A site, whose effect is to change the deleterious effect of G->A mutation into slightly deleterious or neutral. This reviewer thinks in these two scenarios the restorative effect of editing is different: the latter is harm-permitting; while the former is rescue, which might be defined as “adaptive” since it can increase the fitness after the fixation of the deleterious mutations. The authors might consider separate these two scenarios in their model (Fig. 1) and their data analysis.

Response

We agree that not all restorative editing is harm-permitting and did not assume all restorative editing to be harm-permitting. For instance, some nonsynonymous G-to-A substitutions are neutral, so the corresponding restorative editing would be neutral too. That is, only a subset of restorative editing is harm-permitting. This is now clarified in the main text (page 6, paragraph 1).

We agree with the reviewer that the editing in the second scenario described in the comment is harm-permitting and nonadaptive. These cases are classified as restorative editing in our analysis; so there is no concern here.

With regard to the first scenario described in the comment ("rescue"), we also agree with the reviewer that it is adaptive and is not harm-permitting. In this rescue scenario, the G-to-A substitution can occur at any time prior to the emergence of the editing at the site, which most likely took place when the editing

activity was substantially elevated in the branch immediately preceding the common ancestor of coleoids. Because the G-to-A substitution can happen at any time before the elevation of the editing activity, while the most recent common ancestor of cephalopods (i.e., the common ancestor of the top five species in Fig. 2A) was no more than 100 MY older than the elevation, the probability that the substitution occurred in the 100 MY window and hence the editing is classified as restorative is small. In other words, rescue events are unlikely to affect our analysis of restorative editing sites. Although such rescue events are potentially included in diversifying editing sites analyzed, diversifying editing still show lower editing frequencies and editing levels when compared with synonymous editing. Thus, our interpretation that diversifying editing is overall under purifying selection remains valid. Furthermore, even for the minority of rescue events that are classified as restorative, the impact is small. This is because deleterious G-to-A mutations that could get fixed without editing are presumably only slightly deleterious. Hence the benefit of A-to-G editing at such sites is also presumably tiny such that their editing levels may not be selectively raised or selectively maintained at high levels. More importantly, there will be a comparable number of slightly beneficial G-to-A substitutions followed by slightly deleterious A-to-G editing that are included in the category of restorative editing. The effects of these two groups of events are likely cancelled out. We have added the above points to the main text (page 12, paragraph 2).

Comment 3

The dN/dS analysis used by the authors and other previous studies is designed to detect the signature of natural selection on DNA mutations. The editing events can be actually treated as “phenotypes” since they are determined and influenced by cis-regulatory elements. The authors are encouraged to discuss about this point to make their results more influential.

Response

We agree completely that the editing level is a phenotypic trait, so our comparison between synonymous and nonsynonymous editing frequencies and levels are distinct from the commonly used dN/dS analysis of DNA sequences. However, the null hypothesis that synonymous editing levels (or frequencies) equal nonsynonymous editing levels (or frequencies) when there is no selection still holds. We have added a brief discussion on this point (page 3, paragraph 1).

Comment 4

Since the conclusion drawn by Jiang & Zhang is somewhat opposite to that of Liscovitch-Brauer et al. (2017, Cell), to be fair, the authors might consider doing the following analysis.

The authors inferred the ancestral state of the sites only for the nonsynonymous editing, and compare the editing density to the synonymous sites. Is it possible the authors also infer the restorative and diversifying synonymous sites and use those as control in the analysis?

Response

Both restorative and diversifying synonymous editing are presumably neutral. So there is no need to separate synonymous editing into these two categories. Because the number of synonymous editing sites is only ~55% of the number of nonsynonymous editing sites, dividing synonymous editing sites into two categories makes the sample size of each category relatively small, reducing the power of subsequent statistical analyses. More importantly, ancestral sequence inference at synonymous sites is generally less reliable than that at nonsynonymous sites because of the relatively high substitution rates at synonymous sites. In the present case, d_s between octopus and squid is already greater than 1, rendering the ancestral sequence inference at synonymous sites challenging and the downstream classification of synonymous editing into the two categories unreliable. Based on these considerations, we decide not to separate synonymous editing into restorative and diversifying editing and explain it in the main text (page 6, paragraph 3).

Comment 5

The authors analyzed 3587 1-to-1 orthologous genes in the six species. How about the remaining genes? If the authors remove the sites that show signals of restorative editing and do dN/dS analysis at the genome-scale, what would the results look like?

Response

Genes lacking detectable orthologs in some species may have orthologs in these species but the orthologs are undetected for technical reasons such as high sequence divergence. Consequently, not removing restorative editing sites in these genes from the analysis would cause overestimation of F_N and L_N . That is why we did not consider these genes in our analysis.

To examine the impact of including these genes in the analysis, we followed the reviewer's suggestion of analyzing all genes except restorative editing sites from the 3587 one-to-one orthologs. One can see from the results in the following table that F_N is still significantly smaller than F_S in the new analysis, but L_N becomes significantly greater than L_S likely owing to the high editing levels of restorative editing sites from non-one-to-one orthologs. Because this analysis is expected to be biased, we decide not to present it in the manuscript.

Table. Synonymous and nonsynonymous editing in all genes excluding restorative editing from one-to-one orthologs

Species	Editing frequencies		Median editing levels (%)	
	Synonymous (F_S)	Nonsynonymous (F_N)	Synonymous (L_S)	Nonsynonymous (L_N)
Octopus	0.042	0.033****	2.04	2.18****
Bimac	0.021	0.016****	3.11	3.54****
Squid	0.019	0.015****	3.78	4.72****
Cuttlefish	0.030	0.024****	2.18	2.33****

Stars indicate significant differences from synonymous editing (*, $P < 0.05$; **, $P < 0.01$; ***, $P < 0.001$; ****, $P < 0.0001$; chi-squared test for the F_N - F_S comparison and Mann-Whitney U test for the L_N - L_S comparison).

Comment 6

The editing level is not considered in the analysis, can the authors provide similar plot as Fig. 5A and 5B in Liscovitch-Brauer et al. 2017, Cell?

Response

Following the suggestion, we present in the following figure F_R/F_S and F_D/F_S calculated using editing sites in each of five editing level ranges. Both ratios generally increase with the editing level. While F_R/F_S is generally greater than 1, F_D/F_S exceeds 1 only for sites with editing levels $>60\%$. It is important to stress that only a few percent of diversifying editing sites fall into this range of editing level (see the table below). Thus, this finding is consistent with our conclusion that only a small fraction of diversifying editing may be adaptive. On page 10, we estimated that this fraction is between 0.75% and 1.90% in the four coleoids. We have added the above analysis and the associated figures to the manuscript (see Fig. S3, Table S3, and page 7, paragraph 2).

Table. Fraction of editing sites in each editing level range

Species	Type	0-20%	20%-40%	40%-60%	60%-80%	80%-100%
Octopus	Synonymous	95.62%	2.82%	0.95%	0.51%	0.11%
	Restorative	89.78%	5.62%	2.84%	1.75%	0.88%
	Diversifying	93.7%	3.53%	1.29%	1.03%	0.44%
Bimac	Synonymous	94.02%	3.74%	1.58%	0.54%	0.13%
	Restorative	89.62%	4.77%	3.18%	2.43%	0.65%
	Diversifying	91.52%	4.69%	2.04%	1.24%	0.52%
Squid	Synonymous	93.26%	3.98%	1.55%	0.78%	0.42%
	Restorative	86.69%	5.28%	3.96%	4.08%	4.56%
	Diversifying	89.20%	4.43%	2.04%	2.08%	2.24%
Cuttlefish	Synonymous	95.71%	2.73%	1.01%	0.44%	0.11%
	Restorative	88.24%	5.76%	3.29%	2.71%	1.06%
	Diversifying	94.05%	3.04%	1.33%	1.18%	0.39%

Comment 7

The authors detect the signals of adaptation by pooling all the editing sites from different tissues. If the authors analyzed the data for each individual tissue, do the patterns change?

Response

Following the suggestion, we analyzed editing in each bimac tissue with available data. As shown in the table below, the results from combined analysis of all tissues are generally supported by the analysis of individual tissues. Note that after the correction for multiple testing, L_R is still significantly higher than L_S in sub and supra but L_D is no longer significantly higher than L_S in any tissue. This new result is shown in Table S2 and mentioned on page 7, paragraph 2.

Table. Editing frequencies and median editing levels in individual bimac tissues

Tissue	Editing frequencies			Median editing levels (%)		
	Synonymous (F_S)	Restorative (F_R)	Diversifying (F_D)	Synonymous (L_S)	Restorative (L_R)	Diversifying (L_D)
Axial nerve cord (ANC)	0.0039	0.0072****	0.0027****	1.35	1.88	1.92
Optical lobe (OL)	0.0032	0.0062****	0.0023****	1.95	8.10	1.43
Subesophageal brain (sub)	0.0055	0.0095****	0.0038****	2.32	3.56****	2.60*
Supraesophageal brain (supra)	0.0053	0.0093****	0.0036****	1.40	2.15****	1.53**
Posterior salivary gland (PSG)	0.00021	0.00060****	0.00017	0.92	NA	0.16
Skin	0.00034	0.00076****	0.00023****	0.24	0.49	0.23
Sucker	0.00066	0.0015****	0.00051****	0.44	0.39	0.31
Retina	0.00071	0.0018****	0.00051****	0.48	0.37	0.55
Ovary	0.00013	0.00029*	0.0000084*	0.39	NA	0.23
Testes	0.00031	0.00066****	0.00022***	0.21	0.22	0.17
Viscera	0.00027	0.00060****	0.00018***	0.19	0.56	0.20
Stage 15 embryo (ST15)	0.0010	0.0023****	0.00074****	0.56	0.53	0.44

Significant differences from synonymous editing are indicated by stars (*, $P < 0.05$; **, $P < 0.01$; ***, $P < 0.001$; ****, $P < 0.0001$). Editing frequencies in each tissue are calculated using the numbers of potential editing sites in genes expressed in the tissue. Chi-squared test and Mann-Whitney U test are used for comparisons of editing frequencies and editing levels, respectively. As in the main analysis, editing levels are compared for sites covered by at least 400 RNA-seq reads. No restorative editing site satisfying this condition is found in two tissues; their L_R values are shown as “NA”. After the correction for multiple testing, L_R is still significantly higher than L_S in sub and supra, but L_D is no longer significantly higher than L_S in any tissue.

Reviewer #2

Comment 1

In this study, the authors proposed that the widespread nonsynonymous RNA editing in coleoids is generally nonadaptive, which is in contrast to the conclusion of a previous study by Liscovitch-Brauer and colleagues. It is indeed interesting to investigate the evolutionary forces that shape the abundant RNA editing events in coleoids. However, I found that the analysis is not strong enough to allow the authors to draw such a conclusion.

Response

We are glad that the reviewer considers the topic of our study to be interesting. Specific comments are addressed below.

Comment 2

There are not enough details about the parameters used to select genuine RNA editing sites used for analysis. The median editing levels of different groups of sites in several figures (Fig. 2C, Fig. 4B, Fig. S1B) is 1-2% (nearly equal to the sequencing error rate of NGS data), suggesting that most sites the authors used are not even really edited. Any conclusion based on such site lists is unreliable. The authors should require a minimum editing level (e.g. 5%) to select sites for analysis.

Response

Editing sites analyzed in this study are exactly the same set of sites analyzed by Liscovitch-Brauer et al. (2017). Liscovitch-Brauer et al. performed necessary controls and filters to ensure that their calling of editing is reliable. For example, they showed that, compared with the number of A-to-G editing sites called, the numbers of other types of DNA-RNA mismatches were tens to hundreds of times lower (see their Fig. 1). Because sequencing error should not cause tens to hundreds of times more A-to-G error than other types of error, the vast majority of A-to-G editing sites called are genuine. The reviewer's comment of an approximately 1% NGS sequencing error is based on one sequencing read. When tens to hundreds of reads cover a site, as in the case here, the rate of error caused by sequencing is much lower than 1%. For these reasons, we believe that it is both reliable and appropriate to analyze exactly the same set of editing sites as analyzed by the original authors. We have added in Methods (page 16, paragraph 3) an explanation of the set of editing sites used and the rationale of using this set.

For the above reasons, we believe that one need not remove sites with editing levels lower than 5%. Furthermore, because synonymous and nonsynonymous editing sites have different editing level distributions (Fig. 2C), applying the 5% editing level cutoff will create biases. Nevertheless, in response to a different question from Reviewer 1, we compared between synonymous and nonsynonymous editing sites in five different editing level ranges (0-20%, 20-40%, 40-60%, 60-80%, and 80-100%). The results obtained also address this comment and have been incorporated into the revised manuscript. Please see our response to Comment 6 from Reviewer 1.

Comment 3

The authors proposed the harm-permitting model of RNA editing. In this model, they consider a genomic position in a coding region that is occupied by G and does not accept A. As the editing activity in the species rises, a G-to-A mutation at the site may become neutral and fixed by genetic drift if the resultant A is edited back to G in a sufficient proportion of mRNAs. In practice, most editing events are edited in neuronal tissues only and the unedited forms of mRNAs are present in the non-neuronal tissues. Thus the editing positions do accept A and the harm-permitting model is invalid for most of the editing positions.

Response

The reviewer is right that the harm-permitting effect will not work in genes with necessary functions in tissues where they cannot be edited. All of our results are consistent with this premise, and Fig. 3 even shows positive evidence for this premise. In fact, the number of restorative editing sites is ~10% of the number of diversifying editing sites (see Table S1), supporting the reviewer's view. Nevertheless, because restorative editing tends to have high editing levels, it disproportionately contributes to highly edited sites. For example, nearly one half of nonsynonymous editing sites with editing levels higher than 80% belong to the restorative category (Table S3). It is for this reason that restorative editing can explain the seemingly adaptive signal observed by Liscovitch-Brauer et al. among highly edited sites. These points have been incorporated into the manuscript.

Comment 4

Even in the neuronal tissues, the median editing level of restorative sites is only around 2% and most mRNAs are actually unedited (Fig. 2C). So the A allele is also accepted in the neuronal tissues. This observation does not support the harm-permitting model proposed by the authors.

Response

While restorative editing can be harm-permitting, we do not claim that all restorative editing is harm-permitting. For instance, some nonsynonymous G-to-A substitutions are neutral, so the corresponding restorative editing would be neutral too. We have added the above points to the manuscript (page 6, paragraph 1). As a result, the observation that some restorative editing sites have low editing levels does

not contradict the harm-permitting model. What is important is the observation that the median editing level is significantly higher for restorative editing than synonymous editing (Fig. 2C), as predicted by the harm-permitting model. Furthermore, the proportion of sites with high editing levels is greater for restorative editing than other types of editing. For instance, 13.31% of restorative editing sites but only 6.74% of synonymous editing sites have editing levels >20% in the squid (Table S3). The table below shows the percentage of editing sites (with at least 400 reads) for which the editing level exceeds 5%. Depending on the harm of the G-to-A mutation and the relative dominance of the A and G isoforms, 5% of editing from A to G could substantially increase its fixation probability. These points have been added to page 7, paragraph 1.

Table. Percentage of editing sites with editing levels exceeding 5%

	Synonymous	Restorative
Octopus	17.48	28.59****
Bimac	21.33	31.60****
Squid	22.97	33.37****
Cuttlefish	16.74	27.42****

Stars indicate significant differences from synonymous editing (*, $P < 0.05$; **, $P < 0.01$; ***, $P < 0.001$; ****, $P < 0.0001$; chi-squared test).

Comment 5

Fig. 2C: The observed lower editing levels of diversifying sites are likely because these sites are newly evolved and have not been optimized for editing. To allow for a fair comparison between Ld and Ls, a matched evolutionary age between diversifying sites and control synonymous sites is needed. In fact, as shown in Fig. 4B, when using sites with similar evolutionary age, the authors found that Ld is higher than Ls, which supports the adaptive hypothesis.

Response

We agree with the reviewer that an age-matched analysis would strengthen our argument. We have done so in Fig. S1B, where the editing levels of species-specific synonymous, restorative, and diversifying editing sites are compared. The results are very similar to those in Fig. 2C, further supporting our conclusion. That age-matched analyses reveal no adaptive signals in Fig. S1B (species-specific editing) but adaptive signals in Fig. 4B (shared editing) indicate that only shared editing sites contain a sizable fraction of adaptive editing. We have added these results to page 7, paragraph 2.

Comment 6

It is known that editing sites tend to be clustered together. Conserved nonsynonymous sites typically have higher ADAR binding affinity and editing levels, thus the nearby synonymous sites tend to have higher editing levels also. This kind of confounding effect may affect the editing level comparison between synonymous and diversifying sites. So a more rigorous selection of matched synonymous sites is needed throughout the analysis.

Response

Clustered sites can reduce the statistical power of the comparison but should not bias the results. For instance, if editing at a nonsynonymous site is beneficial, *cis*-regulatory mutations that increase the editing level at the site will be selected for. If the mutations happen to raise the editing levels of nearby synonymous editing, the difference between synonymous and nonsynonymous editing levels will be smaller than when such ripple effects are absent. This may decrease the statistical power in detecting the difference between synonymous and nonsynonymous editing, but should not cause spurious results because the ripple effect is blind and does not favor one type of editing over another.

To minimize the impact of the ripple effect suggested by the reviewer, we conducted the following analysis. We rank all genes to be analyzed according to their d_N/d_S ratio (calculated by comparing octopus and squid orthologs). Genes with ranks #1, 3, 5, etc are grouped into bin 1 whereas genes with ranks #2, 4, 6, etc are grouped into bin 2. We then compare synonymous editing from bin 1 with nonsynonymous editing from bin 2 (panels A and C in the figure below), and compare nonsynonymous editing from bin 1 with synonymous editing from bin 2 (panels B and D in the figure below). Because we compare synonymous and nonsynonymous editing sites of different genes, the ripple effect is minimized. It can be seen that the overall patterns are the same as in Fig. 2. Specifically, $F_R > F_S$ and $F_D < F_S$ are observed in all cases except one where F_R and F_S do not differ significantly (cuttlefish in panel B). $L_R > L_S$ and $L_D \approx L_S$ are also observed in most cases, except two where there are small differences between L_D and L_S ($L_D > L_S$ in squid but $L_D < L_S$ in octopus). The new results are presented in the manuscript (see Fig. S2 and page 7, paragraph 2).

Reviewers' Comments:

Reviewer #1:

Remarks to the Author:

The authors have adequately addressed my concerns. Therefore, I would like to recommend acceptance.

Reviewer #2:

Remarks to the Author:

I agree that most of the sites identified by Liscovitch-Brauer et al. are genuine, given that tens to hundreds of times more A-to-G variants than other types of variants were observed. However, there is one more issue with the use of sites with low editing levels. For such editing sites, it is difficult to determine if they are indeed species- or clade- specific, and this may affect some of the results in this study, e.g. figure 4 and figure s1.

In a typical RNA-seq data, most of the genomic positions are covered by <100 reads. Let us assume that a site in species A has a 2% editing level and is covered by 50 or 100 reads (1 or 2 G reads are sequenced here). When examining the orthologous site in species B with the same editing level, given the low editing level and limited read coverage, the G reads may not be able to be sampled and sequenced. Therefore, the authors may need to deduce the editing level and coverage requirement to determine if a site is indeed unedited in a given species.

I still have some concern that the harm-permitting model may not be able to apply to most of the restorative sites, given their low editing levels. For example, as shown in Table S3, 87% of restorative editing sites have editing levels < 20% in the squid.

Responses to the reviewers

Reviewer #1

Comment

The authors have adequately addressed my concerns. Therefore, I would like to recommend acceptance.

Response

We thank the reviewer for reviewing our revised manuscript and for the recommendation of acceptance.

Reviewer #2

Comment 1

I agree that most of the sites identified by Liscovitch-Brauer et al. are genuine, given that tens to hundreds of times more A-to-G variants than other types of variants were observed. However, there is one more issue with the use of sites with low editing levels. For such editing sites, it is difficult to determine if they are indeed species- or clade- specific, and this may affect some of the results in this study, e.g. figure 4 and figure S1.

In a typical RNA-seq data, most of the genomic positions are covered by <100 reads. Let us assume that a site in species A has a 2% editing level and is covered by 50 or 100 reads (1 or 2 G reads are sequenced here). When examining the orthologous site in species B with the same editing level, given the low editing level and limited read coverage, the G reads may not be able to be sampled and sequenced. Therefore, the authors may need to deduce the editing level and coverage requirement to determine if a site is indeed unedited in a given species.

Response

We thank the reviewer for reviewing our revised manuscript. We agree that some so-called species-specific editing may in fact be shared among species; they may have been mistakenly identified as species-specific due to low sequencing coverage and/or low editing level that reduces the detectability of editing. While the group of identified species-specific editing may be contaminated with some shared editing, the group of identified shared editing is not contaminated with species-specific editing. Hence, true signals from species-specific editing can be inferred by comparing the results in Fig. S1 (identified species-specific editing) and Fig. 4 (identified shared editing). Specifically, we infer that the true signals from species-specific editing after the removal of the contamination from shared editing are as follows.

- (1) Because shared editing between two species shows $F_R < F_S$ (Fig. 4A), removal of contamination from shared editing would strengthen the signal of $F_R > F_S$ observed in species-specific editing (Fig. S1). In other words, the true signal of $F_R > F_S$ should be even stronger than what is shown in Fig. S1.
- (2) The true signal of $F_D < F_S$ should be similar to that seen in Fig. S1, because the relationship between F_D and F_S is similar in Fig. 4A and Fig. S1.
- (3) The true signal of $L_R > L_S$ should be similar to that seen in Fig. S1, because the relationship between L_R and L_S is similar in Fig. 4B and Fig. S1.
- (4) Because shared editing between two species shows $L_D > L_S$ (Fig. 4B), removal of contamination from shared editing would strengthen the signal of $L_D < L_S$ observed in species-specific editing (Fig. S1). In other words, the true signal of $L_D < L_S$ should be even stronger than what is shown in Fig. S1.

We briefly summarize the above finding on page 7 of the main text (highlighted in yellow) and refer readers to Fig. S1 legend (highlighted in yellow), where we provide the above information in details.

Comment 2

I still have some concern that the harm-permitting model may not be able to apply to most of the restorative sites, given their low editing levels. For example, as shown in Table S3, 87% of restorative editing sites have editing levels $< 20\%$ in the squid.

Response

We agree with the reviewer that not all restorative editing is necessarily harm-permitting. For instance, some nonsynonymous G-to-A substitutions are neutral, and the corresponding restorative editing would also be neutral. Having a subset of restorative editing that is neutral does not contradict the harm-permitting model or affect its ability to explain the observation of higher nonsynonymous editing levels than synonymous editing levels in coleoids.

In particular, the reviewer asks whether the moderate editing levels (e.g., $< 20\%$) of most restorative editing are sufficient to permit otherwise harmful G-to-A substitutions. We note that the harm-permitting model is proposed as an alternative to the adaptive hypothesis in an attempt to explain the generally higher nonsynonymous editing levels than synonymous editing levels. The adaptive hypothesis must also assume that moderate editing levels (i.e., $< 20\%$) are sufficient to confer advantages detectable by natural selection. In other words, both competing hypotheses must assume that moderate levels of nonsynonymous editing has detectable fitness effects. So, the moderate nonsynonymous editing levels do not make the harm-permitting model disfavored when compared with the adaptive hypothesis. To the contrary, we observed the general trend of $L_R > L_S$ and $L_D < L_S$, which supports the harm-permitting model relative to the adaptive hypothesis. In short, we agree that most nonsynonymous editing in coleoids does not have high editing levels, but even this observation supports the harm-permitting model more than the adaptive hypothesis.

We have added the above discussion to the manuscript (highlighted in yellow on page 7).

Reviewers' Comments:

Reviewer #2:

Remarks to the Author:

The authors have addressed my concerns, and I believe that the manuscript is ready for publication.

Response to reviewers

Reviewer #2

Comment

The authors have addressed my concerns, and I believe that the manuscript is ready for publication.

Response

We thank the reviewer for reviewing our manuscript.